# Human Milk Antibodies after BNT162b2 Vaccination Exhibit Reduced Binding against SARS-CoV-2 Variants of Concern

**DOI:** 10.3390/vaccines10020225

**Published:** 2022-01-31

**Authors:** Jia Ming Low, Yue Gu, Melissa Shu Feng Ng, Liang Wei Wang, Zubair Amin, Youjia Zhong, Paul A. MacAry

**Affiliations:** 1Department of Neonatology, Khoo Teck Puat-National University Children’s Medical Institute, National University Health System, Singapore 119074, Singapore; jia_ming_low@nuhs.edu.sg (J.M.L.); paeza@nus.edu.sg (Z.A.); 2Department of Microbiology and Immunology, Yong Loo Lin School of Medicine, National University of Singapore, Singapore 117585, Singapore; micgy@nus.edu.sg (Y.G.); micpam@nus.edu.sg (P.A.M.); 3Singapore Immunology Network, Agency for Science, Technology and Research, Immunos Building, Singapore 138648, Singapore; Melissa_Ng@immunol.a-star.edu.sg; 4Department of Paediatrics, Yong Loo Lin School of Medicine, National University of Singapore, Singapore 117585, Singapore

**Keywords:** breast milk antibodies, passive immunity, Pfizer vaccination, COVID-19, immunoglobin A, BNT162b2, Comirnaty, tozinameran

## Abstract

SARS-CoV-2-specific antibody responses are engendered in human milk after BNT162b2 vaccination. However, the emergence of variants of concern (VOCs) raises concerns about the specificity of and potential cross-protection mediated by milk antibody responses, which are crucial for passive immunity transferred from breastfeeding mothers to their infants. In this study, we collected milk samples at three different time points pre- and post-vaccination, and measured milk IgA antibody binding to the receptor binding domain (RBD) of the original Wuhan-Hu-1 strain, and the four VOCs, namely Alpha, Beta, Gamma and Delta. We report a significant level of anti-RBD IgA in milk collected at 4–6 weeks after the second dose of vaccination compared to pre-vaccination. We observed around a 30% reduction in binding to most VOCs, including the major circulating Delta variant, compared to the original Wuhan-Hu-1 strain. As COVID-19 vaccines may take some time to be approved for infants, these individuals remain at risk for severe disease and rely mainly on transferred passive immunity. Our findings support the current recommendations for vaccinating lactating women with the aim of transferring mucosal immunity to breastfeeding infants.

## 1. Introduction

SARS-CoV-2-specific antibody responses are engendered in human milk after vaccination with Pfizer/BioNTech’s BNT162b2 (also known as Comirnaty; INN: tozinameran) [1], as well as Moderna’s mRNA-1273 [2]. Both of these mRNA vaccines encode a human-codon-optimized SARS-CoV-2 spike glycoprotein, using the first described Wuhan-Hu-1 strain as the basis for design, together with proline substitutions at lysine 986 and valine 987 to stabilize the prefusion intermediate. However, over the course of the pandemic, numerous variants have arisen, numbering nine as of December 2021 [3]. While some of these variants are of minimal clinical concern and are deemed as variants of interest, World Health Organization (WHO)-designated variants of concern (VOCs) have increased potential for spread and may cause more severe disease. Given that existing vaccines, including BNT162b2, utilize earlier virus strains for their design, questions have been raised regarding the specificity and potential cross-protection mediated by these antibodies. While antibody responses have been extensively studied for vaccine sera [4], human milk antibodies—a major contributor to passive immunity for infants—have not been analyzed. In fact, there is some de-coupling of antibody expression between blood and milk compartments; high virus-specific antibody titers in serum do not necessarily predict similar quantities in milk. Thus, further studies on milk antibodies generated to the COVID-19 vaccines are warranted.

Here, we investigated whether BNT162b2 vaccination induced secretion of specific immunoglobulin A (IgA) into human milk against the principal determinant of neutralization (Spike-receptor binding domain, RBD) for four major VOCs, namely Alpha, Beta, Gamma and Delta. These data illuminate our understanding of transferred mucosal immunity in lactating mothers receiving mRNA vaccines.

## 2. Materials and Methods

### 2.1. Subjects and Milk Sample Collection

We conducted a prospective cohort study of a convenience sample of lactating women in Singapore. Participants were recruited between 5 and 9 February 2021 through advertisements. All received 2 doses of the Pfizer/BioNTech BNT162b2 vaccine, with the second dose (D2) given 21 days after dose 1. All lactating mothers who were more than 21 years old who had COVID-19 vaccines or who were planning to get the COVID-19 vaccines were included. Women with active autoimmune disease, infectious disease, cancer or on any immunomodulatory medication were excluded from the study.

A total of 20 mL of human milk samples was self-collected before administration of vaccine, 3–7 days after D2, and 4–6 weeks after D2 (Figure 1A). Each human milk sample was either hand-expressed or expressed through a breast pump before breastfeeding and was collected at participants’ convenience during the specified time points. We did not collect samples during the inter-dose period, as we had previously established the lack of significant increases in spike-specific antibody titers during that time. We recruited participants through convenience sampling. Sample size calculation was not performed, as this is a cohort study. In light of the ongoing COVID-19 pandemic, it was deemed imperative to report any emerging evidence to allow breastfeeding mothers, physicians and policymakers to make a more informed decision for COVID-19 vaccination in breastfeeding mothers, while larger studies powered to statistical significance were underway. The study was approved by the Institutional Review Board (DSRB 2021/00095); informed consent was obtained. The study was registered at ClinicalTrials.gov, accessed on 10 January 2022 (NCT04802278).

Milk samples were delipidated by centrifugation twice, at 10,000× *g*, at 4 °C, for 15 min each time. The clear portion was stored at −20 °C and thawed before use.

### 2.2. Biochemical Analysis

Receptor binding domain (RBD) of SARS-CoV-2 ancestral Wuhan-Hu-1 strain (WH-1), Alpha (B.1.1.7), Beta (B.1.351), Gamma (P.1) and Delta (B.1.617.2) variants were diluted in PBS and coated on 384-well Maxisorp plates (NUNC) at 80 ng/well for overnight incubation at 4 °C. The plate was washed 3 times with 1x PBST buffer (0.05% Tween in 1x PBS) and blocked with 3% bovine serum albumin in 1x PBST for 1.5 h incubation at room temperature. Delipidated milk was diluted 20-fold in the blocking buffer. After 3 plate washes in 1x PBST, diluted milk samples were added at 20 µL/well for 1 h incubation at room temperature. The plate-wash procedure was repeated thrice. Anti-human F(ab’)2 anti-human IgA-HRP (Invitrogen, #A24458) was diluted 5000 times in the blocking buffer and added to the plate at 20 µL/well for 1 h incubation at room temperature, protected from light. After 3 plate washes in 1x PBST, 1-Step Ultra TMB-ELISA (Thermo Scientific, #34029) was added at 20 µL/well. Reaction was stopped with 20 µL/well 1 M H_2_SO_4_ 3 min later. OD_450_ was measured by using a microplate reader (Tecan Spark). Results were based on 3 technical replicates per sample.

### 2.3. Statistical Analysis

Statistical analysis was performed with R (4.0.2), using the PMCMRPlus package (1.9.0). Tests performed were Kruskal–Wallis, followed by Dunn’s Many-to-One Rank Comparison Test for comparisons of variants to WH-1. The *p*-values less than 0.05 were considered as statistically significant.

## 3. Results

Forty-six subjects (mean age 31.5 years, 13.5 months post-partum) completed the study (Table 1).

IgA antibodies against RBDs of WH-1 strain, and four major VOCs were found in milk within 3–7 days after D2. Compared to WH-1, RBD-specific IgA was significantly reduced by 28–33% for the Beta, Gamma and Delta variants at this time point (Figure 1B). The minor difference detected for the Alpha variant did not achieve statistical significance.

At 4–6 weeks after D2, milk IgA against WH-1 and all VOCs decreased but were still higher than the pre-vaccination baseline. IgA binding relative to WH-1 was reduced for all VOCs, except for Alpha, by 25–30% (Figure 1C).

## 4. Discussion

The presence of vaccine-elicited spike-specific antibodies likely confers some protection to the breastfed infants, who are ineligible for vaccination and are at risk of severe COVID-19 [5].

Although spike-specific IgG is known to be elicited and secreted into breast milk, we did not measure these antibodies, as they have hitherto no firmly established roles in mucosal immunity. Nonetheless, based on previous work, it is expected that spike-specific IgG levels track with that for IgA in vaccinated mothers [1].

It is worth noting that most of our samples were collected from mothers around 13.5 months post-partum and, therefore, represented mature milk. It is not fully known whether the antibody profiles of mRNA-vaccinated mothers evolve over the course of milk maturation from colostrum to transitional to mature milk. However, it is difficult to assess these changes, if any, due to lower take-up rates among pregnant women in Singapore (50% as of August 2021 [6] versus 80% of the general population at that time) [7] that limit sample availability.

This study lends supports to the increasing importance of antenatal vaccination of pregnant women to facilitate transplacental transfer of IgG to the infant as it wanes slower at 5–6 months of life, thereby augmenting both the magnitude and duration of passive immunity transferred from mother to child [8]. Further research on optimizing the antenatal vaccination schedule is needed to maximally exploit the transplacental transmission of passive immunity. Additionally, lactating mothers as a population could be prioritized for booster vaccine doses to maximize transferred immunity to vulnerable infants.

Finally, while the existing vaccines remain effective against serious disease and subsequent hospitalization events, their ability to target emerging variants may not be on par with that for earlier variants [9], which is, in part, supported by our findings with milk-borne antibodies in the present study. To counter the antigenic drift in SARS-CoV-2, future strategies against COVID-19 may entail continual revision of vaccines and world-wide virus surveillance in a manner akin to that for influenza viruses. Another promising approach to avoid entering an evolutionary arms race is a universal vaccine. Recent work by Swadling and colleagues show that abortive SARS-CoV-2 infection without seroconversion can occur in individuals with pre-existing polymerase-specific T cells [10]. Future iterations of the vaccine may, therefore, focus on virus-encoded RNA-dependent RNA polymerase to stimulate sterilizing immunity—one step ahead of neutralizing antibody responses elicited by current vaccines, which are vulnerable to immune escape with variant emergence.

We acknowledge certain limitations of this study. No functional assays were performed; however, SARS-CoV-2-specific IgA binding in vaccine milk has been positively correlated with neutralization [8]. This is a small and relatively homogenous population; larger studies are warranted.

## 5. Conclusions

We detected the presence of SARS-CoV-2-specific IgA against all major VOCs in milk up to 6 weeks after D2 of BNT162b2. However, we also observed significantly reduced binding of these antibodies to VOCs, including the globally dominant Delta variant, suggesting reduced protection for breastfeeding infants. Additionally, these antibodies were significantly reduced by as early as 4–6 weeks after D2 [1].

## Figures and Tables

**Figure 1 vaccines-10-00225-f001:**
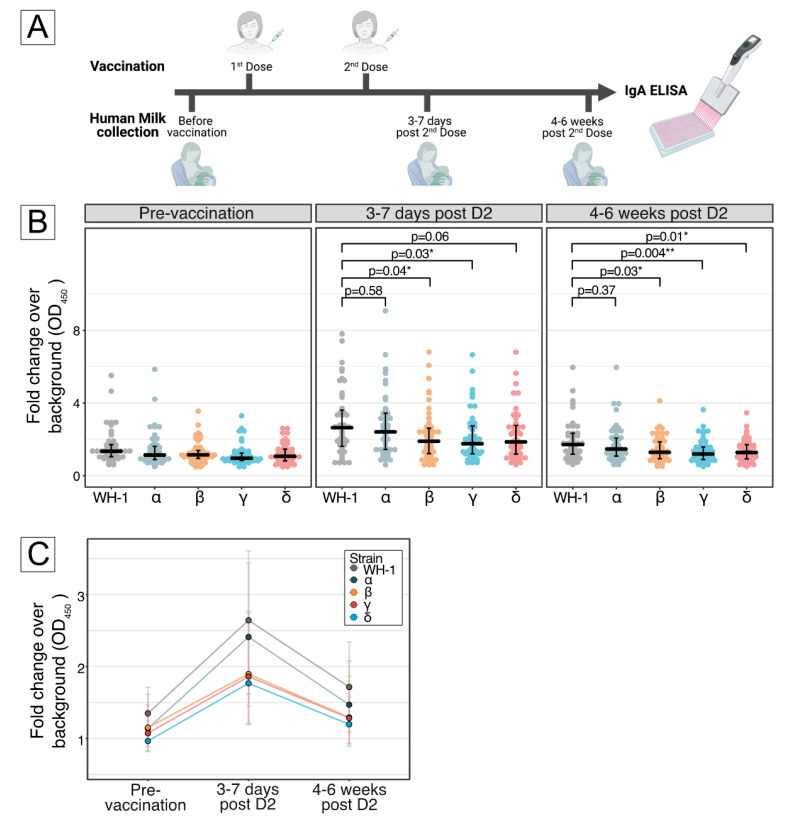
Detection of IgA binding to receptor binding domain of SARS-CoV-2 spike proteins encoded by the ancestral Wuhan-Hu-1 strain and selected major VOCs. (**A**) Schematic showing the collection times of human milk samples pre- and post-vaccination, as described in Materials and Methods. (**B**) Differential human milk IgA binding to receptor binding domain (RBD) of SARS-CoV-2 spike protein of the ancestral Wuhan-Hu-1 strain (WH-1), Alpha (B.1.1.7), Beta (B.1.351), Gamma (P.1) and Delta (B.1.617.2) variants of concern, after vaccination with BNT162b2 across 3 time points: pre-vaccination, 3–7 days post-D2 and 4–6 weeks post-D2. Each dot denotes an individual sample. Center line denotes the median, and error bars show Quartile 1 (bottom bar) and Quartile 3 (top bar). The *p*-values were calculated by using Dunn’s Many-to-One Rank Comparison Test and reported for VOC compared against WH-1, where *p* < 0.05 was considered significant; *p* < 0.05 is marked with *, and *p* < 0.01 is marked with **. At 3–7 days post-D2, reduction from WH-1 for Alpha = 8%, Beta = 28%, Gamma = 33% and Delta = 30%. At 4–6 weeks post-D2, reduction from WH-1 for Alpha = 14%, Beta = 25%, Gamma = 30% and Delta = 25%. (**C**) Kinetics of human milk IgA binding across 3 time points for ancestral Wuhan-Hu-1 strain (WH-1), Alpha (B.1.1.7), Beta (B.1.351), Gamma (P.1) and Delta (B.1.617.2) variants of concern. Each line represents 1 strain. Dots represent the median, and error bars represent interquartile range.

**Table 1 vaccines-10-00225-t001:** Characteristic of lactating women.

**Number of participants**	**No. (%)**
Study participants, No.	46
Maternal characteristics	
Maternal Age, mean (standard deviation, SD), years	31.5 (0.9)
Ethnicity Singaporean Chinese Singaporean Malay	41 (89.1) 5 (10.9)
Chronic disease Asthma Polycystic Ovarian Syndrome Thalassemia Minor	1 (2.2) 1 (2.2) 1 (2.2)
Antenatal conditions	0
Smoker	0
Postpartum (standard deviation, SD), months	13.5 (SD 2.1)

## Data Availability

The data are confidential but can be shared in an anonymized manner upon request.

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
