# Peer review of "Human Milk Antibodies after BNT162b2 Vaccination Exhibit Reduced Binding against SARS-CoV-2 Variants of Concern"

_vaccines, 2022, doi:10.3390/vaccines10020225_

Round 1

Reviewer 1 Report

This work is especially interesting, it is very well described and current. It is explored how the specific IgA of SARS-CoV-2 variants is transferred to the breast milk with mother with full vaccination. They provide very current data on the passive immunity of neonates.

I would like to thank the authors for reporting this data and for being able to review their work. Some considerations to make the article more attractive could be:

  • The introduction is very well written, clear and straightforward. The objective of the study is perfectly understood.
  • In the material and methods, a figure representing the timming for sample collection process may be useful. On the other hand, the sample size, and the criteria for inclusion and exclusion of women should also be described (age, previous complications such as asthma, obesity, cancer, etc., smoking habits, alcohol consumption, etc.) 
  • The volume colllected also needs to be reported, as well as the time of the draw or if it was a 24-hour pool. Was the withdrawal after feeding the neonate?
  • the section on biochemical analysis and statistics should be separated with subsections. On the other hand, the p-value that was used as significant should be reported.
  • The results section is clear and concise, although Table 1 would be included in the material and methods section, the figures are very clarifying.
  • In the discussion section, it might be interesting to include that the mean number of months postpartum was 13.5 and that in the mature milk stage immunity might be compromised, needing to explore how Igs behave in colostrum and transitional milk. On the other hand, I agree with the authors that these are data that support immune protection of the neonate through breastfeeding and that IgG and cell-mediated immunity could be another defense barrier.

  • the conclusions section should be separate

Author Response

Responses to Reviewer’s comments

 Reviewer: 1

Comment1: This work is especially interesting, it is very well described and current. It is explored how the specific IgA of SARS-CoV-2 variants is transferred to the breast milk with mother with full vaccination. They provide very current data on the passive immunity of neonates.

I would like to thank the authors for reporting this data and for being able to review their work. Some considerations to make the article more attractive could be:

Reply: Thank you for the kind words in appreciation of our work. We have revised the manuscript further taking into account all the comments and suggestions as below.

Edits in manuscript:

As in the revised manuscript uploaded in the online portal

 Comment2: The introduction is very well written, clear and straightforward. The objective of the study is perfectly understood.

Reply: Thank you

Edits in manuscript:

Not applicable

  Comment3: In the material and methods, a figure representing the timing for sample collection process may be useful. On the other hand, the sample size, and the criteria for inclusion and exclusion of women should also be described (age, previous complications such as asthma, obesity, cancer, etc., smoking habits, alcohol consumption, etc.) 

Reply: We have included a figure to summarize the sample collection process.

We recruited participants through convenience sampling; we have indicated this in the Materials and Methods section. Sample size calculation was not performed as this is a cohort study. In light of the ongoing COVID-19 pandemic, it was deemed important to report any early emerging evidence to allow breastfeeding mothers, clinicians and policy makers to make a more informed decision for COVID-19 vaccination, while larger studies powered to detect statistical significance were underway. This is also added in the revised manuscript

All lactating mothers more than 21 years old who had COVID-19 vaccines or who were planning to get the COVID-19 vaccines were included. Women with active autoimmune disease, infectious disease, cancer or on any immunomodulatory medication were excluded from the study. We have added these inclusion and exclusion criteria in the Materials and Methods section.

Edits in manuscript:

2.1 Subjects and milk sample collection

“We conducted a prospective cohort study of a convenience sample of lactating women in Singapore. Participants were recruited between February 5–9, 2021 through advertisements. All received 2 doses of the Pfizer/BioNTech BNT162b2 vaccine, with the second dose (D2) given 21 days after dose 1. All lactating mothers more than 21 years old who had COVID-19 vaccines or who were planning to get the COVID-19 vaccines were included. Women with active autoimmune disease, infectious disease, cancer or on any immunomodulatory medication were excluded from the study.”

“20ml of human milk samples were self-collected before administration of vaccine, 3–7 days after D2, and 4-6 weeks after D2 (Figure 1A). Each human milk sample was either hand expressed or expressed through a breast pump before breastfeeding and was collected at participants’ convenience during the specified time points. Figure 1A included in the revised manuscript.”

“We recruited participants through convenience sampling. Sample size calculation was not performed as this is a cohort study. In light of the ongoing COVID-19 pandemic, it was deemed imperative to report any emerging evidence to allow breastfeeding mothers, physicians and policy makers to make a more informed decision for COVID-19 vaccination in breastfeeding mothers, while larger studies powered to statistical significance were underway.”

 Comment4: The volume collected also needs to be reported, as well as the time of the draw or if it was a 24-hour pool. Was the withdrawal after feeding the neonate?

Reply: 20ml of breast milk (hand expressed or expressed through a breast pump) before feeding was collected at each time interval at their convenience at each of the 5 time points; we have added these collection protocol details in the Materials and Methods section.
Edits in manuscript:

2.1 Subjects and milk sample collection

20ml of human milk samples were self-collected before administration of vaccine, 3–7 days after D2, and 4-6 weeks after D2 (Figure 1A). Each human milk sample was either hand expressed or expressed through a breast pump before breastfeeding and was collected at participants’ convenience during the specified time points.”

  Comment5: the section on biochemical analysis and statistics should be separated with subsections. On the other hand, the p-value that was used as significant should be reported.

Reply: We have divided the Materials and Methods into subsections “Subjects and milk sample collection”, “Biochemical analysis” and “Statistical analysis” as suggested. We have also indicated that “p values less than 0.05 were considered as statistically significant” in the subsection “Statistical analysis”.

Edits in manuscript:

Materials and Methods section as included in the revised manuscript.

 Comment6: The results section is clear and concise, although Table 1 would be included in the material and methods section, the figures are very clarifying.

Reply: Thank you for the comment. For now, we have retained Table 1 in Results section since it summarizes the characteristics of the participants, which the STROBE guidelines for cohort studies suggests to be included in Results.

Edits in manuscript:

Not applicable

 Comment7: In the discussion section, it might be interesting to include that the mean number of months postpartum was 13.5 and that in the mature milk stage immunity might be compromised, needing to explore how Igs behave in colostrum and transitional milk. On the other hand, I agree with the authors that these are data that support immune protection of the neonate through breastfeeding and that IgG and cell-mediated immunity could be another defense barrier.

Reply:  We have included additional explanations in the Discussion section with regards to the changes to milk from colostrum to transitional and finally to mature milk.

Edits in manuscript:

Discussion

“It is worth noting that most of our samples were collected from mothers around 13.5 months post-partum and therefore represented mature milk. It is not fully known whether the antibody profiles of mRNA-vaccinated mothers evolve over the course of milk maturation from colostrum to transitional to mature milk. However, it is difficult to assess these changes, if any, due to lower take-up rates among pregnant women in Singapore (50% as of August 2021 [6] versus 80% of the general population at that time) [7] that limit sample availability.”

Comment8: the conclusions section should be separate

Reply: We have separated the Conclusion section as suggested

Edits in manuscript:

Conclusions

“We detected the presence of SARS-CoV-2 specific IgA against all major VOCs in milk up to 6 weeks after D2 of BNT162b2. However, we also observed significantly reduced binding of these antibodies to VOCs, including the globally dominant Delta variant, suggesting reduced protection for breastfeeding infants. Additionally, these antibodies were significantly reduced by as early as 4-6 weeks after D2 [1].”

Reviewer 2 Report

I have reviewed the concise communication entitled “Human Milk Antibodies after BNT162b2 Vaccination Exhibit Reduced Binding against SARS-CoV-2 Variants of Concern” authored by Low, J. M., and cols. Even though it describes the interesting observation regarding the presence of SARS-CoV-2 specific, RBD-directed and of some decreasing breath IgA antibodies in human milk, I found hard to substantiate a discussion of their findings. As such, I can not recommend its publication in its present form.

Major comments

  1. It would be surprising that sIgA directed against the solo vaccination antigen would not be present in breast milk at the analyzed timepoints after immunization. It follows that their recognition profile follows systemic immunity. Discussing about their role in the neonate is speculative and requesting to match the sample with systemic antibody levels is not practical and potentially impossible at this moment. There are two potential areas of interest that can enhance or put in perspective their findings:

A) Given that vaccination happened around one year after delivery, at this moment the breast milk is considered post-mature and their protein content and potentially albumin; globulin ratio different from breast milk at other stages. This is perfectly known and documented in medical literature, thus is doable to document the protein composition of the studied  breast milk samples and compare it to references.

B) There are other immunizations recommended during pregnancy. Depending on the world region, influenza, tetanus or HBV vaccinations might have been received. This is an opportunity to contrast the author’s results with another vaccination antigen and corroborate if IgA is constant or not detectable against another antigen.

Minor comments

  1. Modify the scale  of the axis in figure 1A, it looks like 8-fold change over background is the top value for all timepoints.
  2. Is this a full article or a short communication?

Author Response

Responses to Reviewer’s comments

 Reviewer: 2

 Comment1: I have reviewed the concise communication entitled “Human Milk Antibodies after BNT162b2 Vaccination Exhibit Reduced Binding against SARS-CoV-2 Variants of Concern” authored by Low, J. M., and cols. Even though it describes the interesting observation regarding the presence of SARS-CoV-2 specific, RBD-directed and of some decreasing breath IgA antibodies in human milk, I found hard to substantiate a discussion of their findings. As such, I can not recommend its publication in its present form.

Reply: Thank you for reviewing our manuscript. We have revised the manuscript addressing all the comments.

Edits in manuscript:

All throughout the manuscript, addressing all the comments

 Major comments 

Comment2: 1.It would be surprising that sIgA directed against the solo vaccination antigen would not be present in breast milk at the analyzed timepoints after immunization. It follows that their recognition profile follows systemic immunity. Discussing about their role in the neonate is speculative and requesting to match the sample with systemic antibody levels is not practical and potentially impossible at this moment. There are two potential areas of interest that can enhance or put in perspective their findings:
Reply: We thank the reviewer for the comments. From Figure 1B, we know that secretory IgA is elicited by the vaccine mRNA; its presence is no surprise, since the mRNA vaccine has immunostimulatory properties similar to other conventional vaccines. From our data published in “Low, J.M.; Gu, Y.; Ng, M.S.F.; Amin, Z.; Lee, L.Y.; Ng, Y.P.M.; Shunmuganathan, B.D.O.; Niu, Y.; Gupta, R.; Tambyah, P.A.; et al. Codominant IgG and IgA expression with minimal vaccine mRNA in milk of BNT162b2 vaccinees. NPJ Vaccines 2021, 6, 105, doi:10.1038/s41541-021-00370-z.”, we do know that there is a degree of uncoupling of IgA and IgG expression levels between blood and milk compartments. In other words, vaccinated mothers have different virus-specific IgG: IgA ratios between their blood and milk. Nonetheless, it is fairly well-established that microbe-specific immunoglobulin transferred from mother to infant has a role in neonatal protection, even though the mechanisms are not completely elucidated. We have revised the discussion taking into consideration the above points.

Edits in manuscript:
Discussion

“The presence of vaccine-elicited spike-specific antibodies likely confer some protection to the breastfed infants, who are ineligible for vaccination and are at risk of severe COVID-19 [5].”

“Although spike-specific IgG is known to be elicited and secreted into breast milk, we did not measure these antibodies as they have hitherto no firmly established roles in mucosal immunity. Nonetheless, based on previous work, it is expected that spike-specific IgG levels track with that for IgA in vaccinated mothers [1].”

Comment2: A) Given that vaccination happened around one year after delivery, at this moment the breast milk is considered post-mature and their protein content and potentially albumin; globulin ratio different from breast milk at other stages. This is perfectly known and documented in medical literature, thus is doable to document the protein composition of the studied breast milk samples and compare it to references.

Reply: We thank the reviewer for this suggestion. However, the key point of our manuscript is to show that the vaccine-induced antibodies are less capable of binding newer variants-of-concern, not the evolution of immunoglobulin repertoires over time per se. Furthermore, as our samples were largely collected at the same period postpartum (~1 year), the effects we observed are most likely not related to milk maturation, which typically occurs within the 1st month postpartum. 

Edits in manuscript:

Not applicable

  Comment3: B) There are other immunizations recommended during pregnancy. Depending on the world region, influenza, tetanus or HBV vaccinations might have been received. This is an opportunity to contrast the author’s results with another vaccination antigen and corroborate if IgA is constant or not detectable against another antigen.

Reply: We thank the reviewer for this great suggestion. We agree with your point that majority of our mothers received pertussis and influenza vaccinations as standard of care antenatally. However, we felt that it would not provide a direct counterpoint to the present situation where mothers in our cohort had received the BNT162b2 vaccine about 1 year postpartum hence this was not undertaken in our study.

Edits in manuscript:

Not applicable

  Minor comments 

Comment4: 1.Modify the scale of the axis in figure 1A, it looks like 8-fold change over background is the top value for all timepoints.

Reply: Thank you for the suggestion; we have done so.

Edits in manuscript:

Figure 1B (originally 1A) as included in the revised manuscript

Comment5: 2. Is this a full article or a short communication?

Reply: This manuscript is a short communication submitted under the category of Article.

Edits in manuscript:

Not applicable

Round 2

Reviewer 2 Report

Thank you for your responses.

This manuscript is a resubmission of an earlier submission. The following is a list of the peer review reports and author responses from that submission.